# Gene Expression Signature-Based Approach Identifies Antifungal Drug Ciclopirox As a Novel Inhibitor of HMGA2 in Colorectal Cancer

**DOI:** 10.3390/biom9110688

**Published:** 2019-11-02

**Authors:** Yu-Min Huang, Chia-Hsiung Cheng, Shiow-Lin Pan, Pei-Ming Yang, Ding-Yen Lin, Kuen-Haur Lee

**Affiliations:** 1Department of Surgery, College of Medicine, Taipei Medical University, Taipei 11031, Taiwan; y.m.huang@yahoo.com.tw; 2Division of General Surgery, Department of Surgery, Taipei Medical University Hospital, Taipei 11031, Taiwan; 3Department of Biochemistry and Molecular Cell Biology, School of Medicine, College of Medicine, Taipei Medical University, Taipei 11031, Taiwan; chcheng@tmu.edu.tw; 4Ph.D. Program in Biotechnology Research and Development, College of Pharmacy, Taipei Medical University, Taipei 11031, Taiwan; slpan@tmu.edu.tw; 5Ph.D. Program for Cancer Molecular Biology and Drug Discovery, College of Medical Science and Technology, Taipei Medical University, Taipei 11031, Taiwan; yangpm@tmu.edu.tw (P.-M.Y.); lindy@mail.ncku.edu.tw (D.-Y.L.); 6Graduate Institute of Cancer Biology and Drug Discovery, College of Medical Science and Technology, Taipei Medical University, Taipei 11031, Taiwan; 7Department of Biotechnology and Bioindustry Sciences, College of Bioscience and Biotechnology, National Cheng Kung University, Tainan 003107, Taiwan; 8TMU Research Center of Cancer Translational Medicine, Taipei Medical University, Taipei 11031, Taiwan; 9Cancer Center, Wan Fang Hospital, Taipei Medical University, Taipei 11696, Taiwan

**Keywords:** HMGA2, CPX, gene signature, LINCS L1000

## Abstract

Human high-mobility group A2 (HMGA2) encodes for a non-histone chromatin protein which influences a variety of biological processes, including the cell cycle process, apoptosis, the DNA damage repair process, and epithelial–mesenchymal transition. The accumulated evidence suggests that high expression of HMGA2 is related to tumor progression, poor prognosis, and a poor response to therapy. Thus, HMGA2 is an important molecular target for many types of malignancies. Our recent studies revealed the positive connections between heat shock protein 90 (Hsp90) and HMGA2 and that the Hsp90 inhibitor has therapeutic potential to inhibit HMGA2-triggered tumorigenesis. However, 43% of patients suffered visual disturbances in a phase I trial of the second-generation Hsp90 inhibitor, NVP-AUY922. To identify a specific inhibitor to target HMGA2, the Gene Expression Omnibus (GEO) database and the Library of Integrated Network-based Cellular Signatures (LINCS) L1000platform were both analyzed. We identified the approved small-molecule antifungal agent ciclopirox (CPX) as a novel potential inhibitor of HMGA2. In addition, CPX induces cytotoxicity of colorectal cancer (CRC) cells by induction of cell cycle arrest and apoptosis in vitro and in vivo through direct interaction with the AT-hook motif (a small DNA-binding protein motif) of HMGA2. In conclusion, this study is the first to report that CPX is a novel potential inhibitor of HMGA2 using a drug-repurposing approach, which can provide a potential therapeutic intervention in CRC patients.

## 1. Introduction

Human high-mobility group A2 (HMGA2) is an architectural transcription factor with no transcriptional function. HMGA2 can regulate gene expression, DNA replication, and chromosome repair by binding to the minor groove of DNA through AT-hook motifs [1]. HMGA2 influences a variety of biological processes, including the cell cycle process, apoptosis, the DNA damage repair process, and epithelial–mesenchymal transition [2]. HMGA protein overexpression is associated with neoplastic transformation [3]. In addition, the HMGA2 protein is overexpressed in various types of cancer, including lung cancer [4], ovarian cancer [5], breast cancer [6], oral squamous cell carcinoma [7], pancreatic cancer [8], and colorectal cancer [9]. The accumulated evidence suggests that high expression of HMGA2 is related to tumor progression, poor prognosis, and a poor response to therapy [10,11]. Thus, HMGA2 is an attractive target for many types of cancers.

Human health has been considerably improved through healthcare interventions. However, for some complex diseases such as cancer, there are still no efficient therapies. This is due to the high cost and length of drug development, as well as the fact that existing drugs become less effective because of drug resistance [12]. To overcome the aforementioned costs of drug development in laboratories, in silico approaches have been developed [13]. Microarray analyses of genes enable the measurement of genome-wide expression; it is frequently and extensively used in clinical studies of human cancers [14]. By examining the differential expression of genes compared to normal samples, the gene expression signature of a disease can be created, which can then be used to study perturbations of the biological and physiological systems by small bioactive molecules [15,16,17,18,19].

Our recent studies revealed the positive connections between heat shock protein 90 (Hsp90) and HMGA2, and that the Hsp90 inhibitor (second-generation Hsp90 inhibitor, NVP-AUY922) has therapeutic potential to inhibit HMGA2-triggered tumorigenesis [9]. However, 43% of patients suffered visual disturbances in a phase I trial of NVP-AUY922 [20]. Thus, the use of NVP-AUY922 is limited due to its ocular toxicities. To identify a specific inhibitor to target HMGA2, the Gene Expression Omnibus (GEO) database and Library of Integrated Network-based Cellular Signatures (LINCS) L1000 platform [21] were both analyzed. We identified the approved small-molecule antifungal agent ciclopirox (CPX) as a novel potential inhibitor of HMGA2. The in vitro and in vivo effects of CPX were investigated in colorectal cancer (CRC) cells. Moreover, the direct interaction between HMGA2 and CPX was further investigated.

## 2. Materials and Methods 

### 2.1. Chemicals, Reagents, and Antibodies

Ciclopirox (CPX), crystal violet, methanol, organic solvent, citrate, and the proteasome inhibitor MG132 were obtained from Sigma (St. Louis, MO, USA). Rabbit antibodies against cyclin D1(diluted ratio 1:1000), CDK4 (cyclin-dependent kinase 4) (diluted ratio 1:1000), and poly (ADP-ribose) polymerase (PARP) (diluted ratio 1:1000) were obtained from Cell Signaling (Beverly, MA, USA). HMGA2 (diluted ratio 1:1000) was obtained from Santa Cruz Biotechnology (Santa Cruz, CA, USA). Mouse monoclonal antibodies against caspase-3 (diluted ratio 1:1000) and β-actin (diluted ratio 1:10000) were purchased from Imgenex (San Diego, CA, USA) and MP Biomedicals (Irvine, CA, USA), respectively. The concentrations of secondary antibodies were 1:2000 for all primary antibodies.

### 2.2. Computational Prediction and Assessment of Novel HMGA2 Inhibitor

To perform the computational prediction and assessment of the novel HMGA2 inhibitor, publicly available gene expression measurements were obtained from NCBI GEO [22], with data from HMGA2 overexpression of human colorectal cancer (CRC) DLD-1 cells (GSE136544) and HMGA2 knockdown of human retinoblastoma (RB) Y79 cells (GSE31687). Using the GEO2R online tool (https://www.ncbi.nlm.nih.gov/geo/geo2r/), we created a gene expression signature by deriving the set of differentially expressed genes between the HMGA2 affected and control samples. Next, signature genes were queried using the Library of Integrated Network-based Cellular Signatures (LINCS) L1000 platform [21] to predict which drugs might have the potential to inhibit HMGA2 expression. Drugs with negative enrichment scores have gene expression patterns that are anticorrelated; positive enrichment scores show that they are concordant, and therefore represent putative new therapeutic indications.

### 2.3. Cell Cultures

CRC cell lines were provided by Professor Cheng’s laboratory. All CRC cell lines were cultured in RPMI-1640 (Thermo Fisher Scientific, Waltham, MA, USA), supplemented with 10% fetal bovine serum (FBS) (Thermo Fisher Scientific, Waltham, MA, USA). Antibiotics (Thermo Fisher Scientific, Waltham, MA, USA) were maintained at 37 °C in a humidified atmosphere containing 5% CO_2_.

### 2.4. Cell Viability Assay

Cell viability was determined using the crystal violet staining method, as described previously [23]. In brief, the cells were plated in 96-well cell culture plates at 3000 cells/well and treated with/without CPX at the indicated concentrations. After CPX treatment for 48 h, cells were fixed and stained with 0.5% crystal violet for 10 min at room temperature. Next, the plates were washed three times with water. After drying, the cells were lysed with a 0.1 M sodium citrate buffer (Sigma, St. Louis, MO, USA) and measured at 550 nm using a microplate reader.

### 2.5. Focus Formation Assays

HCT116 colon cancer cells were seeded at 4000 cells in 6-well dishes and grown overnight before treatment with CPX. The medium was changed every three days with CPX. After 11 days, cells were fixed and stained with 0.5% crystal violet. Foci of greater than 5mm in size were counted. All transfections were repeated three times, and average focus counts and standard deviations (SDs) were calculated.

### 2.6. Flow Cytometry Analysis

Cell cycle assays and apoptosis assays were conducted using a flow cytometry-based approach. In order to evaluate the effects of CPX’s influence on cell cycle distribution and/or apoptosis induction, HCT116 cells (2.5 × 10^5^) were treated with CPX for 48 h at the indicated concentration, and then cells were collected in the culture medium, mixed with the Muse cell cycle analysis reagent or annexin V/dead cell reagent, and analyzed using the Muse Cell Analyzer (EMD Millipore, Billerica, MA, USA).

### 2.7. Western Blotting

Cell lines were placed in a lysis buffer at 4 °C for 1 h. The protein samples were analyzed using different percentages of SDS-polyacrylamide gel electrophoresis, as described in [24].

### 2.8. In Vivo Analysis of Effect of CPX

To evaluate the antitumor activity of CPX, human CRC cells (HCT116, 1 × 10^6^, in 0.1 mL serum-free medium) were subcutaneously injected into nude mice. When the tumor sizes reached 200 mm^3^, mice were separated into five treatment groups (four mice in each group) receiving the following treatments: (a) control: 4% ethanol, 5.2% Tween 80, and 5.2% polyethylene glycol 400 vehicle; (b) irinotecan at 100 mg/kg/week; (c) CPX at 25 mg/kg/day; (d) CPX at 50 mg/kg/day; and (e) CPX at 100 mg/kg/day. Tumors were measured weekly using calipers. Tumor size was calculated as volume = (w^2^ × l)/2, where w = width and l = length in mm of the tumor. At the end of the experiments, the animals were sacrificed and the tumors were collected for further analysis. All animal experiments followed ethical standards, and the protocols were reviewed and approved by the Taipei Medical University Institutional Animal Use and Management Committee (approved No.: LAC-2016-0469).

### 2.9. Immunohistochemistry

Tumor tissues were excised and fixed with 10% formalin, embedded in paraffin, and sectioned using a standard histological procedure. Next, the wax was removed with an organic solvent (xylene). The tissue slices attached to the slides were rehydrated, immersed in citrate buffer (pH 6.0), and readied for staining.

### 2.10. In-Silico Modeling Analysis

The docking of CPX to the AT-hook motif of HMGA2 was carried out using SwissDock [25] and analyzed using Chimera [26]. The sequences of the AT-hook motif and DNA duplex were derived from the Protein Data Bank (PDB) (accession code: 3UXW). The binding free energies were averaged to estimate the affinity from the predicted binding intensities of the docking interaction of CPX and the AT-hook motif of HMGA2. The docking parameters were set to default.

### 2.11. Statistical Analysis

Results are presented as the mean ± standard deviations (SDs). Student’s t-test was used to compare all the experiments. Statistical analyses of the cell viability assay and flow cytometry assay were performed using an unpaired Student’s t-test in the Excel software. *P*-values < 0.05 were considered significant.

## 3. Results

### 3.1. Identification of a Potential Inhibitor of HMGA2

To discover a new inhibitor of HMGA2, we analyzed two Gene Expression Omnibus (GEO) datasets deposited into the National Center for Biotechnology Information (NCBI) (Figure 1). First, the gene expression profiles from *HMGA2* overexpression of human colorectal cancer (CRC) DLD-1 cells were derived from the GEO dataset (GSE136544) [27]. Second, the gene expression profiles from *HMGA2* knockdown of human retinoblastoma (RB) Y79 cells were derived from the GEO dataset (GSE31687) [28]. The top 100 differentially expressed genes (100 upregulated and 100 downregulated genes) from the two GEO datasets were analyzed and then queried using the LINCS L1000 platform to predict drugs that might have the potential to inhibit HMGA2 expression. It is well known that overexpression of HMGA is observed in many types of cancer. Therefore, a gene expression signature caused by a drug treatment which is opposite to the gene expression of a disease, indicates that the drug has the potential to treat the disease [29]. Thus, a negative enrichment score indicates that the gene signature of the drug is opposite to the gene signature of the disease, and a positive enrichment score indicates that they are concordant. The top 10 chemical perturbagens with negative enrichment scores for the gene expression signature of overexpression of HMGA2 are listed in Appendix A. Otherwise, the top 10 chemical perturbagens with positive enrichment scores for the gene expression signature of knockdown of HMGA2 are listed in Appendix A. A comparison of these two tables showed that the strongest therapeutic predictions for HMGA2 was associated with Prestwick-1082. However, Prestwick-1082 is a small-molecule perturbation from the Connectivity Map (CMAP) resource (https://portals.broadinstitute.org/cmap/) [29]. To find a positive correlation between the gene expression signature of Prestwick-1082 with the gene expression signature of drugs from the LINCS L1000 platform, the gene expression signature of Prestwick-1082 (https://portals.broadinstitute.org/cmap/) was queried using the LINCS L1000 platform. Among the top 10 chemical perturbagens with positive enrichment scores for the gene expression signature of Prestwick-1082, the first chemical perturbagen was ciclopirox (CPX) (score = 0.805) (Table 1).

### 3.2. CPX Inhibits Cell Proliferation in CRC Cells

To investigate the antiproliferative effect of CPX on CRC cells, four human CRC cell lines were employed. The chemical structure of CPX is shown in Figure 2a. As shown in Figure 2b, CPX inhibited the cell proliferation in a concentration-dependent manner after treatment for 48 h; the IC_50_ values were about 2 μM for the four human CRC cell lines (2.5 μM for HT29, 2.0 μM for HCT116, 1.8 μM for DLD-1, and 2.2 μM for HCT15). To investigate the effects of CPX on the tumorigenic phenotypes of CRC cells, HCT116 was selected and the effects of CPX on colony formation in a focus-forming assay was performed. Strikingly, we found significantly decreased colony numbers and sizes in HCT116 cells after treatment with CPX in a concentration-dependent manner (Figure 2c,d). Taken together, these results support the notion that CPX is a potent anticancer agent in CRC cells.

### 3.3. The Therapeutic Efficacy of CPX in CRC Cells

We next investigated whether CPX-induced cytotoxicity is mediated by cell cycle regulation or apoptotic processes. Propidium iodide (PI) staining, annexin V staining, and western blotting analyses for cell cycle and apoptosis markers were performed. As illustrated in Figure 3a,b, a significant increase in the G1 population was detected in cells treated with CPX at 8 μM for 48 h. In addition, a reduction in the percentages of the S and G2/M population was detected simultaneously for treatment with CPX at 8 μM for 48 h. Next, apoptosis of HCT116 cells was assessed by annexin V staining. As shown in Figure 3c,d, a significant increase in HCT116 cell apoptosis (19.20% ± 2.09% for 4 μM, 30.65% ± 3.32% for 8 μM) (*p* < 0.001) was induced after treatment with CPX for 48 h. Moreover, western blot results revealed that the protein expression of HMGA2 was inhibited after treatment with CPX in a concentration-dependent manner (Figure 3e). Furthermore, there was an increase in the G1 population after CPX treatment as assessed by CDK4 inhibition (Figure 3e). Moreover, the cleavage of caspase-3 and PARP was observed in CPX-treated HCT116 cells (Figure 3e). Overall, these results show that CPX treatment led to G0/G1 cell cycle arrest and apoptosis in CRC cells.

### 3.4. In Vivo Efficacy of CPX in Human CRCHCT116 Xenograft Mice Models

To further evaluate the efficacy of CPX in vivo, ectopic HCT116 tumor xenograft models were established. When the tumor volume was about 200 mm^3^, mice were separated into five treatment groups (four mice in each group) receiving the following treatments: three doses of CPX (25, 50, and 100 mg/kg) daily by oral gavages, a vehicle control group, and a positive control group (irinotecan 100 mg/kg). After 26 days, the CPX (100 mg/kg)-treated tumors presented a 37% tumor growth inhibition relative to the vehicle control (*p* < 0.01) (Figure 4a). In addition, it was noted that after 26 days, the tumor-suppressive response was dose dependent (20%, 31%, and 37% suppression for 25, 50, and 100 mg/kg, respectively; Figure 4a). Moreover, no animals died or showed signs of acute toxicity (maximum weight loss was less than 10%) up to the end of the experimental period (Figure 4b). To determine whether CPX exhibited antitumor efficacy through antiproliferative and apoptosis-inducing effects, at the end of the experimental period (day 26), tumor-bearing mice were sacrificed, then the tumors were excised from the animals and stained with hematoxylin and eosin (H&E). As shown in Figure 4c, CPX (100 mg/kg)-treated mice exhibited visible necrosis cell death effects compared with the control. Furthermore, the protein expression of HMGA2, a cell cycle marker, and apoptosis markers in HCT116-xenografted mice were assessed by a western blot analysis. As illustrated in Figure 4d, the protein expression of HMGA2 and CDK4 was inhibited; the cleavage of caspase-3 and PARP was detected in HCT116-xenografted mice after treatment with CPX at 100 mg/kg (Figure 4d). Collectively, these results indicate that CPX is a potential antitumor agent for CRC.

### 3.5. Molecular Docking of CPX on A-T Hook Motif of HMGA2

The amino acid sequence of the AT-hook motif is a particular feature of HMGA2 [30]. The AT-hook motif is a 10–15-amino-acid-long peptide motif which contains a Gly-Arg-Pro(G-R-P) tripeptide motif and is regarded as the core sequence of this short protein motif [31,32]. In most of the eukaryotes, AT-hooks are involved in regulating chromatin regulatory functions, such as chromatin remodeling and histone modifications [33]. Therefore, the AT-hook motif of HMGA2 plays a crucial role in regulating gene expression by binding to minor grooves of DNA at special AT-rich DNA sequences [34]. To test whether CPX directly interacted with HMGA2, we first retrieved the sequence of the AT-hook motif and the DNA duplex from the Protein Data Bank (PDB) (accession code: 3UXW) [35]. In Figure 5a, we present the complexes of the AT-hook motif and DNA duplex. The left panel is the core sequence of the AT-hook motif as presented in the amino acid sequence of Pro-Arg-Gly-Arg-Pro (P-R-G-R-P). The right panel shows that the AT-hook DNA-binding motifs can bind preferentially to the minor groove of AT-rich DNA. The three-dimensional structure of the complex is shown in Figure 5b, and it is obvious that the Pro-Arg-Gly-Arg-Pro chain of the AT-hook motif is strongly bound to the DNA minor groove. Thus, these results offer evidence in support of our hypothesis that disturbing the binding of the AT-hook motif of HMGA2 to AT-rich DNA sequences in the minor groove of DNA can prevent HMGA2-regulated tumorigenesis. In order to gain further insights into the putative binding mode of CPX with the AT-hook motif of HMGA2, blind docking for CPX with the crystallographic model of the AT-hook motif was carried out. The results obtained from SwissDock [25] were analyzed using Chimera [26] to illustrate the predicted binding mode of CPX’s snug fit into the left pocket of the AT-hook motif. The binding free energies were averaged (−6.56 kcal/mol) to calculate the binding affinity from the predicted binding intensities of the docking interaction of CPX and the AT-hook motif of HMGA2 (Figure 5c). Next, we examined whether downregulation of the HMGA2 protein after CPX treatment due to CPX fitting into the pocket of the AT-hook motif of HMGA2 caused HMGA2 protein instability. As shown in Figure 5d, decreased levels of CDK4 and HMGA2 due to CPX treatment at 1 and 2 μM were stabilized by MG132 (a proteasomal inhibitor) treatment, indicating that proteasomal degradation was involved in the loss of HMGA2 protein expression after CPX treatment. Taken together, these results suggest that downregulation of HMGA2 protein after CPX treatment was through direct interaction between CPX and HMGA2.

## 4. Discussion

The effects of HMGA2 on tumor cell behavior, such as proliferation, invasion, and metastasis, have been reported in many types of cancer [2]. The accumulated evidence indicates that high HMGA2 expression is the preferred therapeutic target for any cancer type [10,11]. This is the first study to identify the clinically used antifungal agent CPX as a novel potential inhibitor of HMGA2. There is a multitude of evidence to support this conclusion. First, we analyzed two GEO datasets including *HMGA2* overexpression and *HMGA2* knockdown gene expression profiles and queried the results using the LINCS L1000 platform to predict whether CPX has the potential to inhibit HMGA2 expression. Second, we found that CPX-induced cytotoxicity is caused by induction of cell cycle arrest and apoptosis in CRC cells. Third, we found that CPX exhibited the antitumor efficacy through antiproliferative and apoptosis-inducing effects in vivo xenograft mice experiments. Fourth, we found that CPX disturbs the binding of the AT-hook motif of HMGA2 to AT-rich DNA sequences in the DNA minor groove to prevent HMGA2-regulated tumorigenesis. Collectively, this is the first report on the study prescribing CPX as a novel potential inhibitor of HMGA2, which can induce cytotoxicity in CRC cells by inducing cell cycle arrest and apoptosis in vitro and in vivo through direct interaction with the AT-hook motif of HMGA2.

CPX, an off-patent antifungal agent, was developed 40 years ago. It has been widely used in the topical treatment of superficial fungal infections [36]. The toxicology of topically applied ciclopirox olamine in humans has been well described; however, the toxicology of CPX administrated orally of is still not clear [37]. Several dose toxicity studies for the oral administration of CPX have been performed in adult animals; for example, three-month repeat-dose toxicity studies were performed in both rats (60 mg/m^2^ per day) and dogs (200 mg/m^2^ per day), which showed no observed adverse effects, demonstrating no toxic effects [38]. Recently, preclinical studies have found that CPX possesses an anticancer effect by inhibiting cell proliferation, inducing cell death, and inhibiting angiogenesis [39,40,41]. Moreover, no evidence that CPX is mutagenic was found after a study involving a single oral gavage dose of CPX in mice. Furthermore, no increase in drug-related neoplasms of CPX was seen compared with control (104-week dermal study in mice) (https://ndclist.com/ndc/51672-1351). In our study, no signs of acute toxicity were observed after oral administration of CPX during the experiment (Figure 4b). Most recently, in a phase I clinical trial study, all patients tolerated oral administration of CPX well at a dose of 40 mg/m^2^ once daily for five days, and disease stabilization and/or hematologic improvement was observed in 2/3 of the advanced hematologic malignancy patients [42]. These findings strongly support the view that there is a great potential concerning the repurposing of CPX for cancer therapy.

Computational approaches have been developed to aid in the discovery of approved drugs through the gene signature approach, which can overcome the high costs and other logistical limitations of experimental high-throughput screening approaches [13,43]. Thus, the repurposing or repositioning of existing drugs for diseases other than what the drugs were approved for may streamline and facilitate pharmaceutical development [44]. Microarray and next generation sequencing (NGS) techniques speed up the generation of genomics data helping to identify the gene signatures for drug-repurposing studies [45]. Hence, a collection of genome-wide transcriptional expression data, from large scale Food and Drug Administration (FDA)-approved small bioactive molecules treated with cultured human cells, were used to study the molecular basis of drug effects [21]. In this study, we identified a specific inhibitor to target HMGA2 by combining the GEO database and the LINCS L1000 platform. Given the importance of this field in drug discovery, the number of publicly available data reporting the effects of drugs on gene expression across diseases is still continuing to grow [44]. Further advances are expected as the availability of chemogenomic data increases and innovations in computation power help to overcome the high costs and other logistical limitations of drug development.

## 5. Conclusions

In conclusion, the results presented in this work provide strong support for the exploration of CPX as a novel potential inhibitor of HMGA2 using the drug-repurposing approach. CPX induced cytotoxicity of CRC cells by inducing cell cycle arrest and apoptosis in vitro and in vivo through direct interaction with the AT-hook motif of HMGA2; this can provide a potential therapeutic intervention in CRC patients.

## Figures and Tables

**Figure 1 biomolecules-09-00688-f001:**
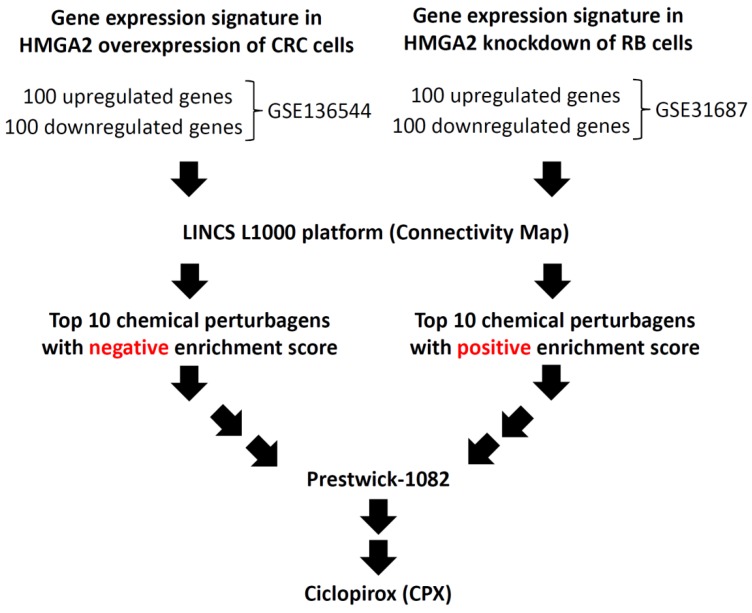
Flow chart of identification of a new inhibitor of HMGA2 using the GEO database and LINCS L1000 platform. The GSE136544 and GSE31687 microarrays were selected from the GEO database. The top 100 differentially expressed genes (100 upregulated and 100 downregulated genes) from the two GEO datasets were analyzed and then queried using the LINCS L1000 platform to predict the drugs that have the potential to inhibit HMGA2 expression. HMGA2—human high-mobility group A2; GEO—Gene Expression Omnibus; LINCS—Library of Integrated Network-based Cellular Signatures; CRC—colorectal cancer; RB—retinoblastoma.

**Figure 2 biomolecules-09-00688-f002:**
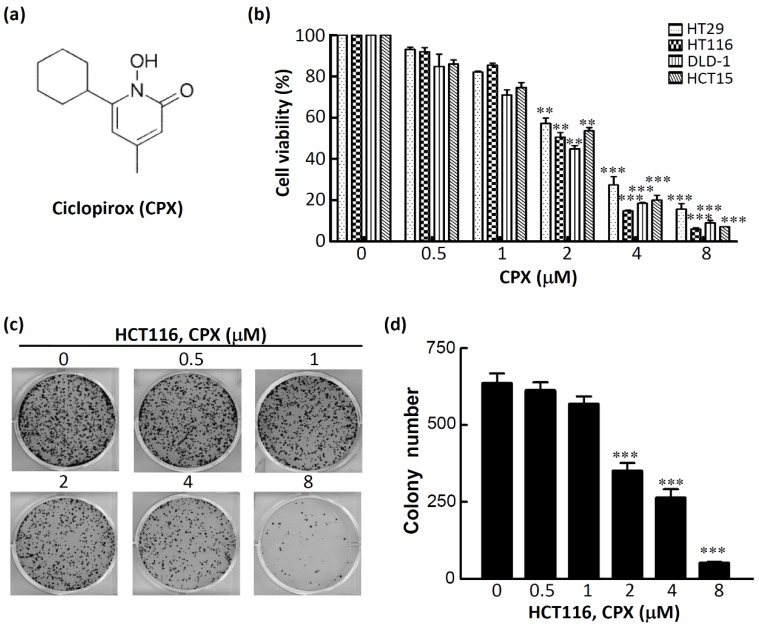
Cytotoxic effect of ciclopirox (CPX) in colorectal cancer (CRC) cell lines. (**a**) Chemical structure of CPX. (**b**) Effects of CPX on the viability of HT29, HCT116, DLD-1, and HCT15 CRC cells. Cells were treated with CPX at the indicated concentrations in 10% fetal bovine serum (FBS)-supplemented medium for 48 h, and cell viability was assessed using the crystal violet staining method. Bars, standard deviation (SD) (*n* = 6). (**c**) HCT116 cells were treated with CPX at the indicated concentrations, and foci were visualized after crystal violet staining. (**d**) The quantification of colony number, wherein total colony counts ± SD are illustrated. ** *p* < 0.01, *** *p* < 0.001.

**Figure 3 biomolecules-09-00688-f003:**
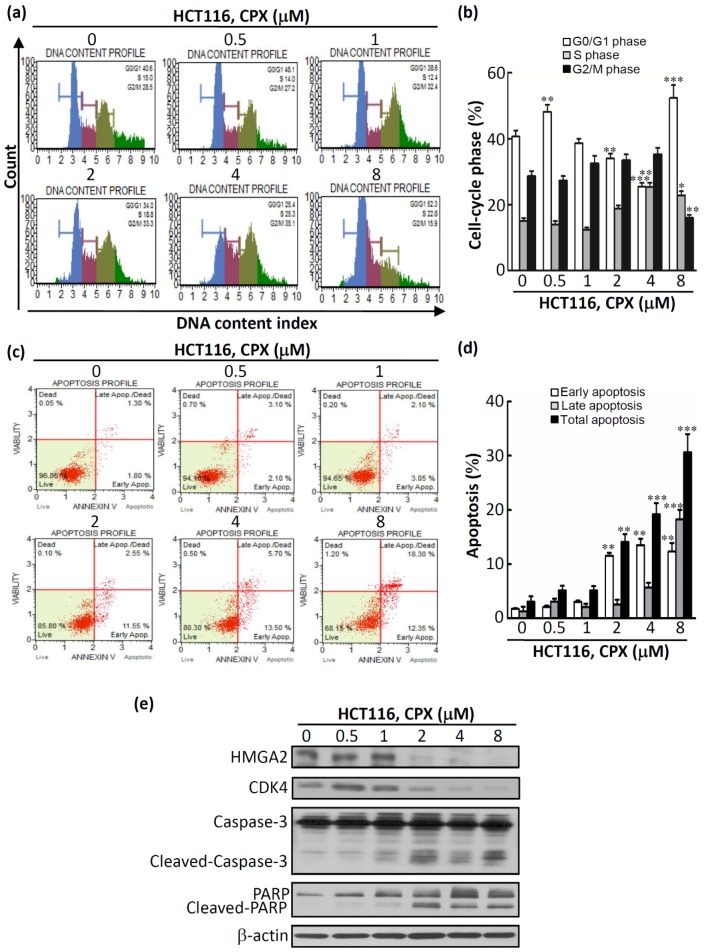
CPX-induced cell cycle arrest and apoptotic death in HCT116 cells. (**a**) HCT116 cells were treated with CPX at the indicated concentrations for 48 h. Cells were subjected to a cell cycle assay using the Muse Cell Analyzer. (**b**) The statistical analysis of cell cycle distribution. (**c**) HCT116 cells were treated with CPX at the indicated concentrations for 48 h. Cells were subjected to an annexin V assay using the Muse Cell Analyzer. The method led to four categories: dead, alive, early apoptosis, and late apoptosis/dead. (**d**) CPX significantly induced total apoptotic cells (black bar) in a dose-dependent manner. (**e**) HCT116 cells were treated with CPX at the indicated concentrations for 48 h, subjected to a western blot analysis, and probed for HMGA2, CDK4, caspase-3, and poly (ADP ribose) polymerase (PARP). β-actin was used as the loading control. * *p* < 0.05, ** *p* < 0.01, *** *p* < 0.001.

**Figure 4 biomolecules-09-00688-f004:**
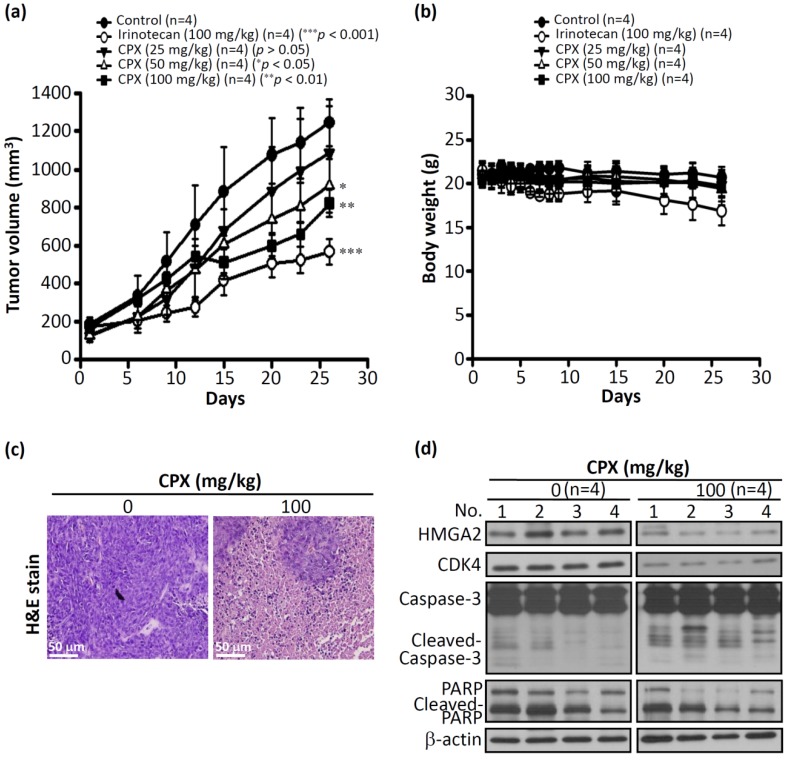
CPX inhibits tumor growth in human CRCHCT116 xenografts in mice. (**a**) Nude mice-derived HCT116 xenograft tumors under indicated treatment: oral gavages with CPX (25, 50, and 100 mg/kg), a vehicle control group, and a positive control group (irinotecan 100 mg/kg), as described in Material and Methods. The tumor volume measurement was performed at indicated times (days). Results are presented as mean± SD (*n* = 4). (**b**) No signs of acute toxicity were observed as assessed by weight loss of less than 10%. (**c**) Tumors were excised from the animals; this was followed by hematoxylin and eosin (H&E) staining. (**d**) Western blot analysis of intratumoral biomarkers of CPX activity in four representative HCT116 tumors from two groups of mice. *p* < 0.05 was considered to be statistically significant. * *p* < 0.05, ** *p* < 0.01, *** *p* < 0.001.

**Figure 5 biomolecules-09-00688-f005:**
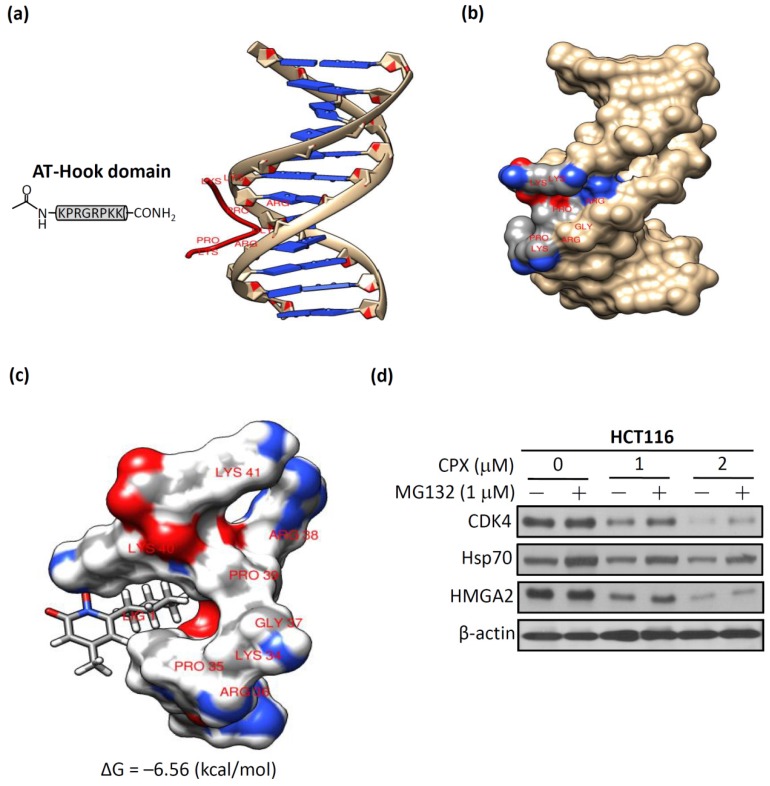
In silico analysis showing docking of CPX to the AT-hook motif of HMGA2 on the crystal structure. (**a**) The complexes of the AT-hook motif and the DNA duplex. (**b**) The Pro-Arg-Gly-Arg-Pro chain of the AT-hook motif is tightly bound to the minor groove of the DNA. (**c**) The binding mode of CPX’s snug fit into the left pocket of the AT-hook motif. The binding free energies were averaged (−6.56 kcal/mol) to estimate the affinity from the predicted binding intensities of the docking interaction of CPX and the AT-hook motif of HMGA2. (**d**) CPX-facilitated suppression of HMGA2 expression was inhibited by treatment with the proteasome inhibitor, MG132 (1 μM, 24 h), in HCT116 cells. HCT116 cells were treated with 1 μM or 2 μM CPX alone for 48 h, or pretreated with CPX for 24 h and combined with MG132 for an additional 24 h, and cell lysates were subjected to a western blot analysis using anti-CDK4, anti-HMGA2, and anti-β-actin antibodies. −: absence of MG132, +: presence of MG132.

**Table 1 biomolecules-09-00688-t001:** Top 10 chemical perturbagens with positive enrichment scores for the gene expression signature of Prestwick-1082.

Rank	Drug	Description	Enrichment Score
1	Ciclopirox	A synthetic antifungal agent for topical dermatologic treatment of superficial mycoses	0.805
2	Tetrandrine	A calcium channel blocker with anti-inflammatory, immunologic, and antiallergenic effects	0.773
3	Viomycin	A group of nonribosomal peptide antibiotics exhibiting anti-tuberculosis properties	0.766
4	Prestwick-692	A small-molecule perturbation from the Connectivity Map (CMAP) resource	0.764
5	Oxyphenbutazone	A nonsteroidal anti-inflammatory drug	0.758
6	Perphenazine	A typical antipsychotic drug	0.753
7	Clotrimazole	A broad spectrum antimycotic or antifungal agent	0.740
8	Metixene	An anticholinergic used as an antiparkinsonian agent	0.739
9	Iloprost	A drug used to treat pulmonary arterial hypertension	0.733
10	Astemizole	A second-generation antihistamine drug	0.732

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
