# Peer review of "Gene Expression Signature-Based Approach Identifies Antifungal Drug Ciclopirox As a Novel Inhibitor of HMGA2 in Colorectal Cancer"

_biomolecules, 2019, doi:10.3390/biom9110688_

Round 1
Reviewer 1 Report
Dear authors
In this study was identified by gene expression the antifungal agent ciclopirox (CPX) as a novel potential inhibitor of HMGA2. CPX induces cell cycle arrest and apoptosis in vitro and in vivo. The paper is clear and the data support the conclusions.
Minors
Please detail the utilized antibodies. In general, be more specific giving more details.
English should be revised e.g. “To understand gene expression signature of Prestwick‑1082 is positive correlation (correlated)…”
Author Response
We appreciate the comments of this reviewer and believe that our manuscript has been improved by attention to him or her. The followings are our responses to the specific issues raised by this reviewer:
Point 1: Please detail the utilized antibodies. In general, be more specific giving more details.
Response 1: We are sorry for the misleading and thank reviewer's valuable suggestion. We have revised Materials and Methods section regarding utilized antibodies in this study in the revised manuscript and shown below:
Rabbit antibodies against Cyclin D1 (diluted ratio 1:1000), CDK4 (diluted ratio 1:1000), and poly(ADP-ribose) polymerase (PARP) (diluted ratio 1:1000) were obtained from Cell Signaling (Beverly, MA, USA). HMGA2 (diluted ratio 1:1000) was obtained from Santa Cruz Biotechnology (Santa Cruz, CA, USA). Mouse monoclonal antibodies against caspase-3 (diluted ratio 1:1000) and β-actin (diluted ratio 1:10000) were purchased from Imgenex (San Diego, CA, USA) and MP Biomedicals (Irvine, CA, USA), respectively. The concentrations of secondary antibodies is 1:2000 for all primary antibodies.
Point 2: English should be revised e.g. “To understand gene expression signature of Prestwick 1082 is positive correlation (correlated)…”
Response 2: We are sorry for the misleading and thank reviewer's valuable suggestion. Our manuscript have sent to MDPI for English editing. The English-editing-certificate shows in PDF file.

Reviewer 2 Report
Dear authors,
nice work, some comments:
1.the grammar and writing of the text must be substantially corrected by someone with fluent English. In some cases the sentence does not make sense
2.line 45: HMGA2 is an architectural transcription factor
3.line 84-58: this sentence was supposed to be explained, but it follows no explanation.
4.line 125: why dit you choose to kill the mouse when the tumor size reached 200mm³. Why not another volume?
5.Enrichment score: it is not clear how did you came up to this score and how it is calculated. Please explain
6.line 136: which "solved" do you mean?
7.lines 138-140: this is not the way how a HE staining is performed or this is not the correct way to describe it. Please correct
8.lines 162-164: this sentence is not clear to me, please explain
9.line 216: what is the definition of the "positive controle group" (it is not clear) and why did you choose irinotecan ?
10.line 217: why did you treat the mouse 26 days? Why not shorter and why not longer?
11.line 219-220: this sentence is not clear to me, please explain
12.line 226: what we see in figure 4c is necrosis and not "antiproliferative effect". This is not a pathological term / diagnosis
13.You discribed that CPX induced cytotoxicity of CRC cells, but did you show any effect in normal cells or structures? Where those afected from the therapy and if yes what was the side effect?
14.You discribed that CPX didnt cause any acute toxicity, but where there any other serious complications in your experiments
15. What is the clinical significance of your results? In Figure 4a I see that irinotecan steel works better than CPX. why should anyone use this treatment?
Author Response
We appreciate the comments of this reviewer and believe that our manuscript has been improved by attention to him or her. The followings are our responses to the specific issues raised by this reviewer:
Point 1: The grammar and writing of the text must be substantially corrected by someone with fluent English. In some cases the sentence does not make sense.
Response 1: We are sorry for the misleading and thank reviewer's valuable suggestion. Our manuscript have sent to MDPI for English editing. The English-editing-certificate please see the attachment.
Point 2: line 45: HMGA2 is an architectural transcription factor.
Response 2: Thanks for reviewer's valuable comments. The sentence has been revised in revised manuscript and shown below:
Human high-mobility group A2 (HMGA2) is an architectural transcription factor with no transcriptional function.
Point 3: line 84-85: this sentence was supposed to be explained, but it follows no explanation.
Response 3: Thanks for reviewer's valuable comments. The sentence has been revised in revised manuscript and shown below:
The computational prediction and assessment of the novel HMGA2 inhibitor was performed as described in the following: Publicly available gene expression measurements were obtained from NCBI GEO, with data from HMGA2 overexpression of human colorectal cancer (CRC) DLD-1 cells (GSE136544) and HMGA2 knockdown of human retinoblastoma (RB) Y79 cells (GSE31687). Using the GEO2R online tool (https://www.ncbi.nlm.nih.gov/geo/geo2r/), we created a gene expression signature by deriving the set of differentially expressed genes between the HMGA2 affected and control samples.
Point 4: line 125: why did you choose to kill the mouse when the tumor size reached 200mm³. Why not another volume?
Response 4: We would like to thank reviewer for the comments. In our study, we described that when the tumor sizes reached 200 mm3, mice were separated into five treatment groups. It has been demonstrated that the tumor sizes reached 200 mm3 of HCT116 tumor xenograft models can start for drug treatment [1].
Reference
[1] Shionome Y, Yan L, Liu S, Saeki T, Ouchi T, Integrity of p53 associated pathways determines induction of apoptosis of tumor cells resistant to Aurora-A kinase inhibitors. PLoS One. 2013;8(1):e55457.
Point 5: Enrichment score: it is not clear how did you came up to this score and how it is calculated. Please explain
Response 5: We thank reviewer’s valuable comments. A value between +1 and -1 representing the relative strength of a given signature in an instance from the total set of instances calculated upon execution of a query. A high positive enrichment score indicates that the corresponding perturbagen induced the expression of the query signature. A high negative enrichment score indicates that the corresponding perturbagen reversed the expression of the query signature. Note that the enrichment score is a relative value, and is a function of the composition of the set of instances upon which the query is executed. The enrichment score differs in this regard from the up scores and down scores, which are absolute values. The absolute strength of a given signature in a given instance can be gauged by the magnitude of the corresponding up score and down score. The enrichment score is a combination of the up score and down score [1].
Reference
[1] Lamb J, Crawford ED, Peck D, Modell JW, Blat IC, Wrobel MJ, Lerner J, Brunet JP, Subramanian A, Ross KN, Reich M, Hieronymus H, Wei G, Armstrong SA, Haggarty SJ, Clemons PA, Wei R, Carr SA, Lander ES, Golub TR, The Connectivity Map: using gene-expression signatures to connect small molecules, genes, and disease. Science. 2006; 313(5795): 1929-35.
Point 6: line 136: which "solved" do you mean?
Response 6: Thanks for reviewer's valuable comments. The wax was removed with a xylene organic solvent.
Point 7: lines 138-140: this is not the way how a HE staining is performed or this is not the correct way to describe it. Please correct
Response 7: We are sorry for the misleading and thank reviewer's valuable suggestion. The sentence has been revised in revised manuscript and shown below:
Tumor tissues were excised and fixed with 10% formalin, embedded in paraffin, and sectioned using a standard histological procedure. Next, the wax was removed with an organic solvent, next, the tissue slices attached to the slides were rehydrated, and immersed in citrate buffer (pH 6.0) and readied for staining.
Point 8: lines 162-164: this sentence is not clear to me, please explain
Response 8: We are sorry for the misleading. The sentence has been revised in revised manuscript and shown below:
Therefore, a gene expression signature caused by drug treatment which is opposite to the gene expression of the disease, indicates that the drug has the potential to treat the disease.
Point 9: line 216: what is the definition of the "positive controle group" (it is not clear) and why did you choose irinotecan ?
Response 9: We are sorry for the misleading and thank reviewer's valuable suggestion. The sentence has been revised in revised manuscript. Irinotecan is used as a standard chemotherapy drug for colorectal cancer (CRC). It has been demonstrated that HCT116 tumor xenograft model respond to irinotecan treatment [1]. Thus, we choose irinotecan to treat with HCT116 tumor xenograft model as the positive control group.
Reference
[1] Motwani M, Jung C, Sirotnak FM, She Y, Shah MA, Gonen M, Schwartz GK, Augmentation of apoptosis and tumor regression by flavopiridol in the presence of CPT-11 in Hct116 colon cancer monolayers and xenografts. Clin Cancer Res. 2001; 7(12): 4209-19.
Point 10: line 217: why did you treat the mouse 26 days? Why not shorter and why not longer?
Response 10: We would like to thank reviewer for the comments. After 26 days, the CPX (100 mg/kg)-treated tumors presented the significant tumor growth inhibition relative to the vehicle control (p<0.01). In addition, it was noted that after 26 days, the tumor-suppressive response was dose dependent (20%, 31%, and 37% suppression for 25, 50, and 100 mg/kg, respectively).
Point 11: line 219-220: this sentence is not clear to me, please explain
Response 11: We are sorry for the misleading. The sentence has been revised in revised manuscript and shown below:
In addition, it was noted that after 26 days, the tumor-suppressive response was dose dependent (20%, 31%, and 37% suppression for 25, 50, and 100 mg/kg, respectively; Figure 4A).
Point 12: line 226: what we see in figure 4c is necrosis and not "antiproliferative effect". This is not a pathological term / diagnosis
Response 12: We are sorry for the misleading and thank reviewer's valuable suggestion. The sentence has been revised in revised manuscript and shown below:
As shown in Figure 4C, CPX (100 mg/kg)-treated mice exhibited visible necrosis cell death effects compared with the control.
Point 13: You described that CPX induced cytotoxicity of CRC cells, but did you show any effect in normal cells or structures? Where those affected from the therapy and if yes what was the side effect?
Response 13: We would like to thank reviewer for the comments. We did not show the effect of CPX in normal cells, but no animals died or showed signs of acute toxicity (maximum weight loss was less than 10%) up to the end of the experimental period. In addition, several dose toxicity studies in adult animals for the oral administration of CPX have been performed, for example, for both rats (60 mg/m2 per day) and dogs (200 mg/m2 per day) in three-month repeat-dose toxicity studies; these show no observed adverse effect, demonstrating no toxic effects [1]. In a phase I clinical trial study, all patients tolerated oral administration of CPX well at a dose of 40 mg/m2 once daily for five days with no toxic effects, and disease stabilization and/or hematologic was improved in 2/3 of the advanced hematologic malignancy patients [2].
Reference
[1] Kellner, H. M.; Arnold, C.; Christ, O. E.; Eckert, H. G.; Herok, J.; Hornke, I.; Rupp, W., Pharmacokinetics and biotransformation of the antimycotic drug ciclopiroxolamine in animals and man after topical and systemic administration. Arzneimittelforschung 1981, 31, (8A), 1337-53.
[2] Minden, M. D.; Hogge, D. E.; Weir, S. J.; Kasper, J.; Webster, D. A.; Patton, L.; Jitkova, Y.; Hurren, R.; Gronda, M.; Goard, C. A.; Rajewski, L. G.; Haslam, J. L.; Heppert, K. E.; Schorno, K.; Chang, H.; Brandwein, J. M.; Gupta, V.; Schuh, A. C.; Trudel, S.; Yee, K. W.; Reed, G. A.; Schimmer, A. D., Oral ciclopirox olamine displays biological activity in a phase I study in patients with advanced hematologic malignancies. Am J Hematol 2014, 89, (4), 363-8.
Point 14: You discribed that CPX didnt cause any acute toxicity, but where there any other serious complications in your experiments
Response 14: We would like to thank reviewer for the comments. In our experiments, no signs of acute toxicity or no any other serious complications were observed after oral administration of CPX during the experiment. In addition, there is no any evidence to show the increase in drug-related neoplasms of CPX was seen compared with control (104-week dermal study in mice) (https://ndclist.com/ndc/51672-1351). Furthermore, It has been demonstrated that oral CPX administered for up to 45 days alleviated clinical symptoms of congenital erythropoietic porphyria (CEP), without signs of toxicity [1].
Reference
[1] Urquiza P, Laín A, Sanz-Parra A, Moreno J, Bernardo-Seisdedos G, Dubus P, González E, Gutiérrez-de-Juan V, García S, Eraña H, San Juan I, Macías I, Ben Bdira F, Pluta P, Ortega G, Oyarzábal J, González-Muñiz R, Rodríguez-Cuesta J, Anguita J, Díez E, Blouin JM, de Verneuil H, Mato JM, Richard E, Falcón-Pérez JM, Castilla J, Millet O., Repurposing ciclopirox as a pharmacological chaperone in a model of congenital erythropoietic porphyria. Sci Transl Med. 2018; 10(459).
Point 15: What is the clinical significance of your results? In Figure 4a I see that irinotecan steel works better than CPX. why should anyone use this treatment?
Response 15: We would like to thank reviewer for the comments. In this study, we identified the specific inhibitor to target HMGA2 by combining the GEO database and the LINCS L1000 platform; this can provide potential therapeutic intervention in CRC patients. In addition, given the importance of this field in drug discovery, this approach can help to overcome the high costs and other logistical limitations of drug development. Irinotecan is used as a standard chemotherapy drug for CRC patients, but, irinotecan has many acute adverse effects. The most prominent and dose limiting being diarrhoea and neutropenia. Diarrhea is a common side effect with many of these drugs, but can be particularly bad with irinotecan. With irinotecan monotherapy, diarrhoea was seen in 80% of patients and severe grade 3 to 4 diarrhoea occurred in 30-40% of the patients [1]. This phenomena can also observe in our study to show in Fig. 4B, to indicate that the body weight lost of mice was more than 10% in irinotecan-treated group, but not in CPX-treated group. Thus, CPX may provide potential therapeutic intervention in CRC patients in the near future.
Reference
[1] Glimelius B, Benefit-risk assessment of irinotecan in advanced colorectal cancer. Drug Saf. 2005;28(5):417-33.
